# Emergence of Language with Multi-agent Games: Learning to Communicate with Sequences of Symbols

**Serhii Havrylov**
ILCC, School of Informatics
University of Edinburgh
s.havrylov@inf.ed.ac.uk

**Ivan Titov**
ILCC, School of Informatics
University of Edinburgh
ILLC, University of Amsterdam
ititov@inf.ed.ac.uk

## Abstract

Learning to communicate through interaction, rather than relying on explicit supervision, is often considered a prerequisite for developing a general AI. We study a setting where two agents engage in playing a referential game and, from scratch, develop a communication protocol necessary to succeed in this game. Unlike previous work, we require that messages they exchange, both at train and test time, are in the form of a language (i.e. sequences of discrete symbols). We compare a reinforcement learning approach and one using a differentiable relaxation (straight-through Gumbel-softmax estimator (Jang et al., 2017)) and observe that the latter is much faster to converge and it results in more effective protocols. Interestingly, we also observe that the protocol we induce by optimizing the communication success exhibits a degree of compositionality and variability (i.e. the same information can be phrased in different ways), both properties characteristic of natural languages. As the ultimate goal is to ensure that communication is accomplished in natural language, we also perform experiments where we inject prior information about natural language into our model and study properties of the resulting protocol.

## 1   Introduction

With the rapid advances in machine learning in recent years, the goal of enabling intelligent agents to communicate with each other and with humans is turning from a hot topic of philosophical debates into a practical engineering problem. It is believed that supervised learning alone is not going to provide a solution to this challenge (Mikolov et al., 2015). Moreover, even learning natural language from an interaction between humans and an agent may not be the most efficient and scalable approach. These considerations, as well as desire to achieve a better understanding of principles guiding evolution and emergence of natural languages (Nowak and Krakauer, 1999; Brighton, 2002), have motivated previous research into setups where agents invent a communication protocol which lets them succeed in a given collaborative task (Batali, 1998; Kirby, 2002; Steels, 2005; Baronchelli et al., 2006). For an extensive overview of earlier work in this area, we refer the reader to Kirby (2002) and Wagner et al. (2003).

We continue this line of research and specifically consider a setting where the collaborative task is a game. Neural network models have been shown to be able to successfully induce a communication protocol for this setting (Lazaridou et al., 2017; Jorge et al., 2016; Foerster et al., 2016; Sukhbaatar et al., 2016). One important difference with these previous approaches is that we assume that messages exchanged between the agents are variable-length strings of symbols rather than atomic categories (as in the previous work). Our protocol would have properties more similar to natural language and, as such, would have more advantages over using atomic categories. For example, it can support compositionality (Werning et al., 2011) and provide an easy way to regulate the amount of

information conveyed in a message. Interestingly, in our experiments, we also find that agents develop a protocol faster when we allow them to use longer sequences of symbols. Somewhat surprisingly, we observe that the language derived by our method favours multiple encodings of the same information, reminiscent of synonyms or paraphrases in natural languages. Moreover, with messages being strings of symbols (i.e. words), it is now possible to inject supervision to ensure that the invented protocol is close enough to a natural language and, thus, potentially interpretable by humans.

In our experiments, we focus on a *referential* game (Lewis, 1969), where the goal for one agent is to explain which image the other agent should select. Our setting can be formulated as follows:

1. There is a collection of images $\{i_n\}_{n=1}^N$ from which a target image $t$ is sampled as well as $K$ distracting images $\{d_k\}_{k=1}^K$.
2. There are two agents: a sender $S_\phi$ and a receiver $R_\theta$.
3. After seeing the target image $t$, the sender has to come up with a message $m_t$, which is represented by a sequence of symbols from the vocabulary $V$ of a size $|V|$. The maximum possible length of a sequence is $L$.
4. Given the message $m_t$ and a set of images, which consists of distracting images and the target image, the goal of the receiver is to identify the target image correctly.

This setting is inspired by Lazaridou et al. (2017) but there are important differences: for example, we use sequences rather than single symbols, and our sender, unlike theirs, does not have access to distracting images. This makes our setting both arguably more realistic and more challenging from the learning perspective.

Generating message $m_t$ requires sampling from categorical distributions over vocabulary, which makes backpropagating the error through the message impossible. It is tempting to formulate this game as a reinforcement learning problem. However, the number of possible messages[1] is proportional to $|V|^L$. Therefore, naïve Monte Carlo methods will give very high-variance estimates of the gradients which makes the learning process harder. Also, in this setup, because the receiver $R_\theta$ tries to adapt to the produced messages it will correspond to the non-stationary environment in which sender $S_\phi$ acts making the learning problem even more challenging. Instead, we propose an effective approach where we use straight-through Gumbel-softmax estimators (Jang et al., 2017; Bengio et al., 2013) allowing for end-to-end differentiation, despite using only discrete messages in training. We demonstrate that this approach is much more effective than the reinforcement learning framework employed in previous approaches to referential games, both in terms of convergence times and the resulting communication success.

Our main contributions can be summarized as follows:

- we are the first to show that structured protocols (i.e. strings of symbols) can be induced from scratch by optimizing reward in collaborative tasks;

- we demonstrate that relaxations based on straight-through estimators are more effective than reinforcement learning for our task;

- we show that the induced protocol implements hierarchical encoding scheme and there exist multiple paraphrases that encode the same semantic content.

## 2 Model

### 2.1 Agents' architectures

The sender and the receiver are implemented as LSTM networks (Hochreiter and Schmidhuber, 1997). Figure 1 shows the sketch of model architecture where diamond-shaped, dashed and solid arrows represent sampling, copying and deterministic functions respectively. The inputs to the sender are target image $t$ and the special token <S> that denotes the start of a message. Given these inputs, the sender generates next token $w_i$ in a sequence by sampling from the categorical distribution $\texttt{Cat}(p_i^t)$ where $p_i^t = \texttt{softmax}(Wh_i^s + b)$. Here, $h_i^s$ is the hidden state of sender's LSTM and can be calculated as[2] $h_i^s = \texttt{LSTM}(h_{i-1}^s, w_{i-1})$. In the first time step we have $h_0^s = \eta(f(t))$ where $\eta(\cdot)$ is an

affine transformation of image features $f(\cdot)$ extracted from a convolutional neural network (CNN). Message $m_t$ is obtained by sequentially sampling until the maximum possible length $L$ is reached or the special token `<S>` is generated.

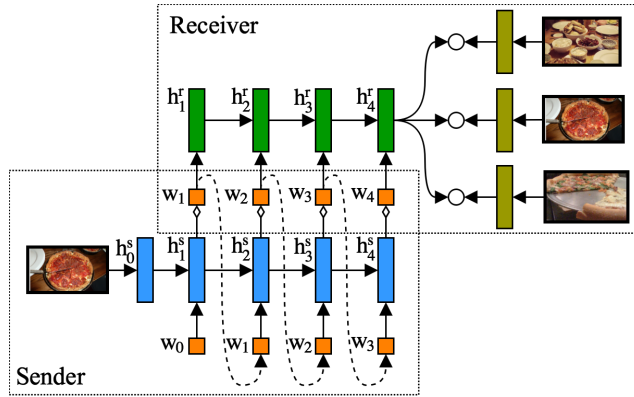

Figure 1: Architectures of sender and receiver.

The inputs to the receiver are the generated message $m_t$ and a set of images that contain the target image $t$ and distracting images $\{d_k\}_{k=1}^{K}$. Receiver interpretation of the message is given by the affine transformation $g(\cdot)$ of the last hidden state $h_l^r$ of the LSTM network that reads the message. The loss function for the whole system can be written as:

$$\mathcal{L}_{\phi,\theta}(t) = \mathbb{E}_{m_t \sim p_\phi(\cdot|t)} \left[ \sum_{k=1}^{K} \max[0, 1 - f(t)^T g(h_l^r) + f(d_k)^T g(h_l^r)] \right] \tag{1}$$

The energy function $E(v, m_t) = -f(v)^T g(h_l^r(m_t))$ can be used to define the probability distribution over a set of images $p(v|m_t) \propto e^{-E(v,m_t)}$. Communication between two agents is successful if the target image has the highest probability according to this distribution.

## 2.2 Grounding in Natural Language

To ensure that communication is accomplished with a language that is understandable by humans, we should favour protocols that resemble, in some respect, a natural language. Also, we would like to check whether using sequences with statistical properties similar to those of a natural language would be beneficial for communication. There are at least two ways how to do this.

The indirect supervision can be implemented by using the Kullback-Leibler (KL) divergence regularization $D_{KL}(q_\phi(m|t)\|p_{NL}(m))$, from the natural language to the learned protocol. As we do not have access to $p_{NL}(m)$, we train a language model $p_\omega$ using available samples (i.e. texts) and approximate the original KL divergence with $D_{KL}(q_\phi(m|t)\|p_\omega(m))$. We estimated the gradient of the divergent with respect to the $\phi$ parameters by applying ST-GS estimator to the Monte Carlo approximation calculated with one sampled message from $q_\phi(m|t)$. This regularization provides indirect supervision by encouraging generated messages to have a high probability in natural language but at the same time maintaining high entropy for the communication protocol. Note that this is a weak form of grounding, as it does not force agents to preserve 'meanings' of words: the same word can refer to a very different concept in the induced artificial language and in the natural language.

The described indirect grounding of the artificial language in a natural language can be interpreted as a particular instantiation of a variational autoencoder (VAE) (Kingma and Welling, 2014). There are no gold standard messages for images. Thus, a message can be treated as a variable-length sequence of discrete latent variables. On the other hand, image representations are always given. Hence they are equivalent to the observed variable in the VAE framework. The trained language model $p_\omega(m)$ serves as a prior over latent variables. The receiver agent is analogous to the generative part of the VAE, although, it uses a slightly different loss for the reconstruction error (hinge loss instead of log-likelihood). The sender agent is equivalent to an inference network used to approximate the posteriors in VAEs.

Minimizing the KL divergence from the natural language distribution to the learned protocol distribution can ensure that statistical properties of the messages are similar to those of natural language. However, words are not likely to preserve their original meaning (e.g. the word 'red' may not refer to 'red' in the protocol). To address this issue, a more direct form of supervision can be considered. For example, additionally training the sender on the image captioning task (Vinyals et al., 2015), assuming that there is a correct and most informative way to describe an image.

## 2.3   Learning

It is relatively easy to learn the receiver agent. It is end-to-end differentiable, so gradients of the loss function with respect to its parameters can be estimated efficiently. The receiver-type model was investigated before by Chrupała et al. (2015) and known as Imaginet. It was used to learn visually grounded representations of language from coupled textual and visual input. The real challenge is to learn the sender agent. Its computational graph contains sampling, which makes it nondifferentiable. In what follows in this section, we discuss methods for estimating gradients of the loss function in Equation (1).

### 2.3.1   REINFORCE

REINFORCE is a likelihood-ratio method (Williams, 1992) that provides a simple way of estimating gradients of the loss function with respect to parameters of the stochastic policy. We are interested in optimizing the loss function from Equation (1). The REINFORCE algorithm enables the use of gradient-based optimization methods by estimating gradients as:

$$\frac{\partial \mathcal{L}_{\phi,\theta}}{\partial \phi} = \mathbb{E}_{p_\phi(\cdot|t)} \left[ l(m_t) \frac{\partial \log p_\phi(m_t|t)}{\partial \phi} \right] \tag{2}$$

Where $l(m_t)$ is the learning signal, the inner part of the expectation in Equation (1). However, computing the gradient precisely may not be feasible due to the enormous number of message configurations. Usually, a Monte Carlo approximation of the expectation is used. Training models with REINFORCE can be difficult, due to the high variance of the estimator. We observed more reliable learning when using stabilizing techniques proposed by Mnih and Gregor (2014). Namely, we use a baseline, defined as a moving average of the reward, to control variance of the estimator; this results in centering the learning signal $l(m_t)$. We also use a variance-based adaptation of the learning rate that consists of dividing the learning rate by a running estimate of the reward standard deviation. This trick ensures that the learning signal is approximately unit variance, making the learning process less sensitive to dramatic and non-monotonic changes in the centered learning signal. To take into account varying difficulty of describing different images, we use input-dependent baseline implemented as a neural network with two hidden layers.

### 2.3.2   Gumbel-softmax estimator

In the typical RL task formulation, an acting agent does not have access to the complete environment specification, or, even if it does, the environment is non-differentiable. Thus, in our setup, an agent that was trained by any REINFORCE-like algorithm would underuse available information about the environment. As a solution, we consider replacement of one-hot encoded symbols $w \in V$ sampled from a categorical distribution with a continuous relaxation $\tilde{w}$ obtained from the Gumbel-softmax distribution (Jang et al., 2017; Maddison et al., 2017).

Consider a categorical distribution with event probabilities $p_1, p_2, ..., p_K$, the Gumbel-softmax trick proceeds as follows: obtain $K$ samples $\{u_k\}_{k=1}^{K}$ from uniformly distributed variable $u \sim U(0, 1)$, transform each sample with function $g_k = -\log(-\log(u_k))$ to get samples from the Gumbel distribution, then compute a continuous relaxation:

$$\tilde{w}_k = \frac{\exp\left((\log p_k + g_k)/\tau\right)}{\sum_{i=1}^{K} \exp\left((\log p_i + g_i)/\tau\right)} \tag{3}$$

Where $\tau$ is the temperature that controls accuracy of the approximation $\arg\max$ with `softmax` function. As the temperature $\tau$ is approaching 0, samples from the Gumbel-softmax distribution

are becoming one-hot encoded, and the Gumbel-softmax distribution starts to be identical to the categorical distribution (Jang et al., 2017).

As a result of this relaxation, the game becomes completely differentiable and can be trained using the backpropagation algorithm. However, communicating with real values allows the sender to encode much more information into a message compared to using a discrete one and is unrealistic if our ultimate goal is communication in natural language. Also, due to the recurrent nature of the receiver agent, using discrete tokens during test time can lead to completely different dynamics compared to the training time which uses continuous tokens. This manifests itself in a large gap between training and testing performance (up to 20% drop in the communication success rate in our experiments).

### 2.3.3 Straight-through Gumbel-softmax estimator

To prevent the issues mentioned above, we discretize $\tilde{w}$ back with $\arg\max$ in the forward pass that then becomes an ordinary sample from the original categorical distribution. Nevertheless, we use continuous relaxation in the backward pass, effectively assuming $\frac{\partial L}{\partial w} \approx \frac{\partial L}{\partial \tilde{w}}$. This biased estimator is known as the straight-through Gumbel-softmax (ST-GS) estimator (Jang et al., 2017; Bengio et al., 2013). As a result of applying this trick, there is no difference in message usage during training and testing stages, which contrasts with previous differentiable frameworks for learning communication protocols (Foerster et al., 2016).

Because of using ST-GS, the forward pass does not depend on the temperature. However, it still affects the gradient values during the backward pass. As discussed before, low values for $\tau$ provide better approximations of $\arg\max$. Because the derivative of $\arg\max$ is 0 everywhere except at the boundary of state changes, a more accurate approximation would lead to the severe vanishing gradient problem. Nonetheless, with ST-GS we can afford to use large values for $\tau$, which would usually lead to faster learning. In order to reduce the burden of performing extensive hyperparameter search for the temperature, similarly to Gulcehre et al. (2017), we consider learning the inverse-temperature with a multilayer perceptron:

$$\frac{1}{\tau(h_i^s)} = \log(1 + \exp(w_\tau^T h_i^s)) + \tau_0, \tag{4}$$

where $\tau_0$ controls maximum possible value for the temperature. In our experiments, we found that learning process is not very sensitive to the hyperparameter as long as $\tau_0$ is less than 1.0.

Despite the fact that ST-GS estimator is computationally efficient, it is biased. To understand how reliable the provided direction is, one can check whether it can be regarded as a pseudogradient (for the results see Section 3.1). The direction $\delta$ is a pseudogradient of $J(u)$ if the condition $\delta^T \nabla J(u) > 0$ is satisfied. Polyak and Tsypkin (1973) have shown that, given certain assumptions about the learning rate, a very broad class of pseudogradient methods converge to the critical point of function $J$. To examine whether the direction provided by ST-GS is a pseudogradient, we used a stochastic perturbation gradient estimator that can approximate a dot product between arbitrary direction $\delta$ in the parameter space and the true gradient:

$$\frac{J(u + \epsilon\delta) - J(u - \epsilon\delta)}{2\epsilon} = \delta^T \nabla J(u) + O(\epsilon^2) \tag{5}$$

In our case $J(u)$ is a Monte Carlo approximation of Equation (1). In order to reduce the variance in dot product estimation (Bhatnagar et al., 2012), the same Gumbel noise samples can be used for evaluating forward and backward perturbations of $J(u)$.

## 3 Experiments

### 3.1 Tabula rasa communication

We used the Microsoft COCO dataset (Chen et al., 2015) as a source of images. Prior to training, we randomly selected 10% of the images from the MSCOCO 2014 training set as validation data and kept the rest as training data. As a result of this split, more than 74k images were used for training and more than 8k images for validation. To evaluate the learned communication protocol, we used the MSCOCO 2014 validation set that consists of more than 40k images. In our experiments

images are represented by outputs of the `relu7` layer from the pretrained 16-layer VGG convolutional network (Simonyan and Zisserman, 2015).

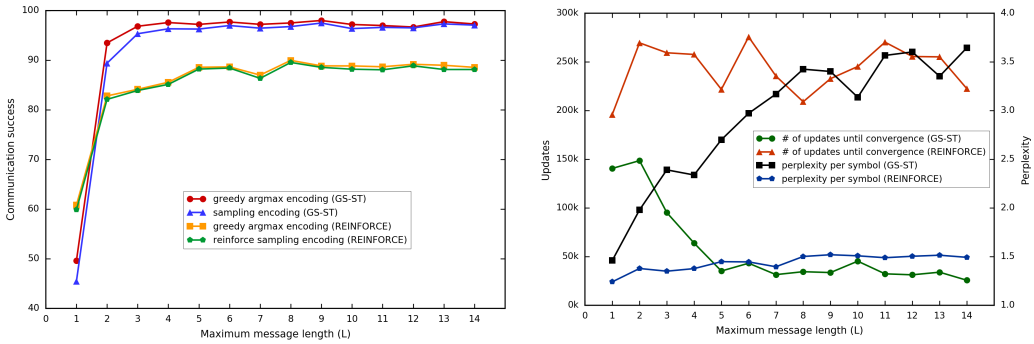

Figure 2: The performance and properties of learned protocols.

We set the following model configuration without tuning: the embedding dimensionality is 256, the dimensionality of LSTM layers is 512, the vocabulary size is 10000, the number of distracting images is 127, the batch size is 128. We used Adam (Kingma and Ba, 2014) as an optimizer, with default hyperparameters and the learning rate of 0.001 for the GS-ST method. For the REINFORCE estimator we tuned learning rate by searching for the optimal value over $[10^{-5}; 0.1]$ interval with a multiplicative step size $10^{-1}$. We did not observe significant improvements while using input-dependent baseline and disregarded them for the sake of simplicity. To investigate benefits of learning temperature, first, we found the optimal temperature that is equal to $1.2$ by performing a search over interval $[0.5; 2.0]$ with the step size equal to $0.1$. As we mentioned before, the learning process with temperature defined by Equation (4) is not very sensitive to $\tau_0$ hyperparameter. Nevertheless, we conducted hyperparameter search over interval $[0.0; 2.0]$ with step size $0.1$ and found that model $\tau_0 = 0.2$ has the best performance. The differences in the performance were not significant unless the $\tau_0$ was bigger than $1.0$.

After training models we tested two encoding strategies: plain sampling and greedy argmax. That means selecting an argmax of the corresponding categorical distribution at each time step. Figure 2 shows the communication success rate as a function of the maximum message length $L$. Because results for models with learned temperature are very similar to the counterparts with fixed (manually tuned) temperatures, we omitted them from the figure for clarity. However, in average, models with learned temperatures outperform vanilla versions by $0.8\%$. As expected, argmax encoding slightly but consistently outperforms the sampling strategy. Surprisingly, REINFORCE beats GS-ST for the setup with $L = 1$. We may speculate that in this relatively easy setting being unbiased (as REINFORCE) is more important than having a low variance (as GS-ST).

Interestingly, the number of updates that are required to achieve training convergence with the GS-ST estimator decreases when we let the sender use longer messages (i.e. for larger $L$). This behaviour is slightly surprising as one could expect that it is harder to learn the protocol when the space of messages is larger. In other words, using longer sequences helps to learn a communication protocol faster. However, this is not at all the case for the REINFORCE estimator: it usually takes five-fold more updates to converge compared to GS-ST, and also there is no clear dependency between the number of updates needed to converge and the maximum possible length of a message.

We also plot the perplexity of the encoder. It is relatively high and increasing with sentence length for GS-ST, whereas for REINFORCE the perplexity increase is not as rapid. This implies redundancy in the encodings: there exist multiple paraphrases that encode the same semantic content. A noteworthy feature of GS-ST with learned temperature is that perplexity values of all encoders for different $L$ are always smaller than corresponding values for vanilla GS-ST.

Lastly, we calculated an estimate of the dot product between the true gradient of the loss function and the direction provided by GS-ST estimator using Equation (5). We found that after 400 parameter updates there is almost always ($> 99\%$) an acute angle between the two. This suggests that GS-ST gradient can be used as a pseudogradient for our referential game problem.

## 3.2 Qualitative analysis of the learned language

To better understand the nature of the learned language, we inspected a small subset of sentences that were produced by the model with maximum possible message length equal to 5. To avoid cherry picking images, we use the following strategy in both food and animal domains. First, we took a random photo of an object and generated a message. Then we iterated over the dataset and randomly selected images with messages that share prefixes of 1, 2 and 3 symbols with the given message. Figure 3 shows some samples from the MSCOCO 2014 validation set that correspond to (5747 ∗ ∗ ∗ ∗) code.[3] Images in this subset depict animals. On the other hand, it seems that images for (∗ ∗ ∗ 5747 ∗) code do not correspond to any predefined category. This suggests that word order is crucial in the developed language. Particularly, word 5747 on the first position encodes presence of an animal in the image. The same figure shows that message (5747 5747 7125 ∗ ∗) corresponds to a particular type of bears. This suggests that the developed language implements some kind of hierarchical coding. This is interesting by itself because the model was not constrained explicitly to use any hierarchical encoding scheme. Presumably, this can help the model efficiently describe unseen images. Nevertheless, natural language uses other principles to ensure compositionality. The model shows similar behaviour for images in the food domain.

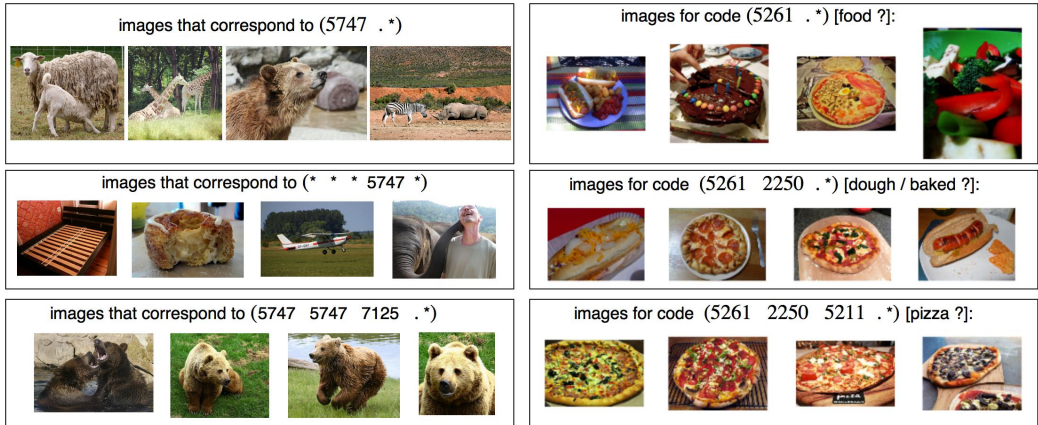

Figure 3: The samples from MS COCO that correspond to particular codes.

## 3.3 Indirect grounding of artificial language in natural language

We implemented indirect grounding algorithm, as discussed in Section 2.2. We trained language model $p_\omega(m)$ using an LSTM recurrent neural network. It was used as a prior distribution over the messages. To acquire data for estimating the parameters of a language model, we took image captions of randomly selected (50%) images from the previously created training set. These images were not used for training the sender and the receiver. Another half of the set was used for training agents. We evaluated the learned communication protocol on the MSCOCO 2014 validation set.

To get an estimate of communication success when using natural language, we trained the receiver with pairs of images and captions. This model is similar to Imaginet (Chrupała et al., 2015). Also, inspired by their analysis, we report the *omission score*. The omission score of a word is equal to difference between the target image probability given the original message and the probability given a message with the removed word. The sentence omission score is the maximum over all word omission scores in the given sentence. The score quantifies the change in the target image probability after removing the most important word. Natural languages have content words that name objects (i.e. nouns) and encode their qualities (e.g., adjectives). One can expect that a protocol that uses a distinction between content words and function words would have a higher omission score than a protocol that distributes information evenly across tokens. As Table 1 shows, the grounded language has the communication success rate similar to natural language. However, it has a slightly lower omission score. The unregularized model has the lowest omission score which probably means that symbols in the developed protocol have similar nature to characters or syllables rather than words.

Table 1: Comparison of the grounded protocol with the natural language and the artificial language

| Model | Comm. success (%) | Number of updates | Omission score |
|---|---|---|---|
| With KL regularization | 52.51 | 11600 | 0.258 |
| Without regularization | 95.65 | 27600 | 0.193 |
| Imaginet | 52.51 | 16100 | 0.287 |

## 3.4 Direct grounding of artificial language in natural language

As we discussed previously in Section 2.2, minimizing the KL divergence will ensure that statistical properties of the protocol are going to be similar to those of natural language. However, words are not likely to preserve their original meaning (e.g. the word 'red' may refer to the concept of 'blue' in the protocol). To resolve this issue, we additionally trained the sender on the image captioning task. To understand whether the additional communication loss can help in the setting where the amount of the data is limited we considered next setup for image description generation task.

To simulate the semi-supervised setting, we divided the previously created training set into two parts. The randomly selected 25% of the dataset were used to train the sender on the image captioning task $\mathcal{L}_{caption}$. The rest 75% were used to train the sender and the receiver to solve the referential game $\mathcal{L}_{game}$. The final loss is a weighted sum of losses for the two tasks $\mathcal{L} = \mathcal{L}_{caption} + \lambda\mathcal{L}_{game}$. We did not perform any preprocessing of the gold standard captions apart from lowercasing. It is important to mention that in this setup the communication loss is equivalent to the variational lower bound of mutual information (Barber and Agakov, 2003) of image features and the corresponding caption.

Table 2: Metrics for image captioning models with and without communication loss

| Model | BLEU-2 | BLEU-3 | BLEU-4 | ROUGE-L | CIDEr | Avg. length |
|---|---|---|---|---|---|---|
| w/ comm. loss | 0.435 | 0.290 | 0.195 | 0.492 | 0.590 | 13.93 |
| w/o comm. loss | 0.436 | 0.290 | 0.195 | 0.491 | 0.594 | 12.85 |

We used the greedy decoding strategy to sample image descriptions. As Table 2 shows, both systems have comparable performance across different image captioning metrics. We believe that the model did not achieve better peroformance as discriminative captions are different in nature compared to reference captions. In fact generating discriminative descriptions may be useful for certain applications (e.g., generating reference expressions in navigation instructions (Byron et al., 2009)) but it is hard to evaluate them intrinsically. Note that using the communication loss yield, in average, longer captions. It is not surprising, taking into account the mutual information interpretation of the referential game, a longer sequence can retain more information about image features.

## 4 Related work

There is a long history of work on language emergence in multi-agent systems (Kirby, 2002; Wagner et al., 2003; Steels, 2005; Nolfi and Mirolli, 2009; Golland et al., 2010). The recent generation relied on deep learning techniques. More specifically, Foerster et al. (2016) proposed a differentiable inter-agent learning (DIAL) framework where it was used to solve puzzles in a multi-agent setting. The agents in their work were allowed to communicate by sending one-bit messages. Jorge et al. (2016) adopted DIAL to solve the interactive image search task with two agents participating in the task. These actors successfully developed a language consisting of one-hot encoded atomic symbols. By contrast, Lazaridou et al. (2017) applied the policy gradient method to learn agents that are involved in a referential game. Unlike us, they used atomic symbols rather than sequences of tokens.

Learning dialogue systems for collaborative activities between machine and human were previously considered by Lemon et al. (2002). Usually, they are represented by hybrid models that combine reinforcement learning with supervised learning (Henderson et al., 2008; Schatzmann et al., 2006).

The idea of using the Gumbel-softmax distribution for learning language in a multi-agent environment was concurrently considered by Mordatch and Abbeel (2017). They studied a simulated

two-dimensional environment in continuous space and discrete time with several agents where, in addition to performing physical actions, agents can also utter verbal communication symbols at every timestep. Similarly to us, the induced language exhibits compositional structure and to a large degree interpretable. Das et al. (2017), also in concurrent work, investigated a cooperative 'image guessing' game with two agents communicating in natural language. They use the policy gradient method for learning, hence their framework can benefit from the approach proposed in this paper. One important difference with our approach is that they pretrain their model on an available dialog dataset. By contrast, we induce the communication protocol from scratch.

VAE-based approaches that use sequences of discrete latent variables were studied recently by Miao and Blunsom (2016) and Kočiskỳ et al. (2016) for text summarization and semantic parsing, correspondingly. The variational lower bound for these models involves expectation with respect to the distribution over sequences of symbols, so the learning strategy proposed here may be beneficial in their applications.

## 5 Conclusion

In this paper, we have shown that agents, modeled using neural networks, can successfully invent a language that consists of sequences of discrete tokens. Despite the common belief that it is hard to train such models, we proposed an efficient learning strategy that relies on the straight-through Gumbel-softmax estimator. We have performed analysis of the learned language and corresponding learning dynamics. We have also considered two methods for injecting prior knowledge about natural language. In the future work, we would like to extend this approach to modelling practical dialogs. The 'game' can be played between two agents rather than an agent and a human while human interpretability would be ensured by integrating supervised loss into the learning objective (as we did in section 3.5 where we used captions). Hopefully, this will reduce the amount of necessary human supervision.

### Acknowledgments

This project is supported by SAP ICN, ERC Starting Grant BroadSem (678254) and NWO Vidi Grant (639.022.518). We would like to thank Jelle Zuidema and anonymous reviewers for their helpful suggestions and comments.

## Footnotes

[1] In our experiments $|V| = 10000$ and $L$ is up to 14.

[2] We omitted the cell state in the equation for brevity.

[3]∗ means any word from the vocabulary or end-of-sentence padding.

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
