[Reviews · NeurIPS 2017]

Reviewer 1



Increasing my score based on the authors rebuttal. The argument that the proposed method can complement human-bot training makes sense. Also, it seems RL baseline experiments were exhaustive. But the argument about the learnt language being compositional should be toned down since there is not enough evidence to support it. Old reviews: The paper proposes to use Gumbel-softmax for training sender and receiver agents in a referential game like Lazaridou (2016). Unlike the previous work, large number of distraction images and variable length messages are allowed. The results suggest the proposed method outperforms Reinforce baseline. In addition, the paper proposes two way to force the communication to be more like natural language. My first concern is that the paper disregards continuous communication for being unlike natural language, and insists on using discrete symbols. But during training, continuous communication is allowed transmit gradients between the agents. This use of back-propagation makes it impossible to learn from interactions with human, which is probably vital for learning natural language. Secondly, I think baseline experiments are not exhaustive and can be improved. Here are my points: - L109: Learnt baseline (network itself predicts baseline values) is pretty standard in deep RL works for it is efficiency over moving average. Why it hasn't tried? Also it is common to add entropy regularization term to the loss to encourage exploration in deep RL. Has it tried? - Appropriate baseline here continuous communication during training and discretization during testing as done in (Foerster, 2016) - L135: a simple solution would be to decrease the temperature to zero during training. Has it tried? - L163: why not hyper-parameters are not tuned? At least the learning rate should be tuned, especially in Reinforce baseline because the default values are tuned for supervised training. At last, the claim of compositionally (L11) of the learnt communication is not supported by any evidence. Also, L196 suggests that word 5747 represents animals. Then, it should still represent the same thing even at different position as in natural language word "animal". The fact that it is not suggest that the learnt communication is actually unlike natural language. Why not similar analysis on reinforce baseline? Other comments: - L59: It is not a non-stationary environment. Since the sender and the receiver have the same reward, they can be considered as a single RL agent. Then, this agent writes words into the environment as "sender", then observes them at the next step as "receiver". - How KL is optimized in sec 3.3? It can't be computed directly because there are exponentially many "m"s. - Why there are two reinforce versions in figure 2 left. What is greedy argmax reinforce? - It would be good to include a histogram of message lengths - Why such large vocabulary size? What is the effect of smaller vocabulary?

Reviewer 2



The authors present a sender/receiver model as an encoder/decoder based on stacked LSTM's and trained straight through with Gumbel-softmax estimations to allow variable-length messages from sender to receiver. The sender encodes information about a target image, which the receiver decodes to select one image out over a hundred distracting images. The paper is well-written and the methodology proposed is novel and works well against a reinforcement-learning-based approach baseline. The qualitative experiments seemed a little unstructured, and the paper wandered for a while in section 3 before wrapping up. Specific comments: -In the abstract, you claim that yours is the first work to require sequences of discrete symbols, but Mordatch and Abbeel also have this requirement (just by time step, rather than one utterance forming to completion before being sent to a receiving agent). -Main contribution's third bullet in the Introduction is vague. This would be a good place to foreshadow the minor, semi-supervised follow-up experiments' results. -Figure 1's caption should have the text explaining what the various arrows mean, not the body of the text. Also, adding captions for image "t", images "d1", "d2", "d3" and the start symbol "< s >" would be helpful to connect the figure to the text. -Why not use the penultimate FC-4096 layer of the VGG network, as is more common practice, to represent images? -I found parts of section 3.1 hard to follow. Did you do your hyperparameter settings based on performance on the validation data you selected from the MSCOCO training data, or on the MCOCO validation data you use as testing data? Which datasets were Figure 2 trained and tested on? -Also in 3.1, it would be good to redefine "success rate" here as the accuracy of picking the correct image, since it could be something like a top-k measure for where the correct image landed among distracting images. -In 3.2, you note as the last sentence that the methodology you describe was used to find the "animal" word as well. Restructuring this section to talk about the selection methodology first will make the whole thing seem stronger and less like cherry-picking. -I couldn't follow section 3.3's experiment well. In Table 1, the final row "Imaginet" seems from the caption to be "the natural language," but from the body of the text I could not figure out what this method represented. -Section 3.5 doesn't contribute much, as a negative result, and could become a paragraph discussing the possibility and how it didn't work, probably. -Section 4 seemed out of place, and could have served as motivation for trusting ST-GS before the experiments were carried out.

Reviewer 3



In this paper, two agents are trained to “communicate” using variable-length strings. The sender encodes a target image as a sequence of symbols. The receiver selects the target image from a set of images based on the symbols. The models are trained using REINFORCE or straight-through Gumbel-softmax, and evaluated on MSCOCO images. Additional experiments attempt to make the symbols closer to natural language. This is an interesting idea, but it might not be ready for publication yet. Some comments: - This model basically produces an image embedding (as a sequence of symbols) which is discriminative enough to select the image out of a batch of 128 images. The motivation is similarity to human language, but it's unclear that any similarities exist. It might be useful to compute some statistics of the learned sequences and compare them to captions. - The experiments in generating a more “natural” language are a bit hard to interpret: -- In the first experiment, the sender is regularized using KL divergence to a language model (trained on image captions). The communication performance drops, and is equal to that of just the receiver trained on captions. It’s hard to conclude much from this. Perhaps the authors could vary the regularization scale and show a trade-off between “naturalness” and discriminative ability of the representation. -- In the second experiment, the sender is trained to both produce captions and communicate with the receiver. This does not make a difference on captioning performance (compared to a model trained just to produce captions), and communication performance is not discussed. It’s again hard to get much insight from this experiment. - The paper would benefit from better organization; e.g. sections on the VAE interpretation and ST-GS as a pseudo-gradient seem a bit out of place.